# Trans-Olecranon Fracture-Dislocations of the Elbow: A Systematic Review

**DOI:** 10.3390/diagnostics10121058

**Published:** 2020-12-06

**Authors:** Chul-Hyun Cho, Du-Han Kim, Sang Soo Na, Byung-Chan Choi, Beom-Soo Kim

**Affiliations:** Department of Orthopedic Surgery, Keimyung University Dongsan Hospital, Keimyung University School of Medicine, 1035, Dalgubeol-daero, Dalseo-gu, Daegu 42601, Korea; oscho5362@dsmc.or.kr (C.-H.C.); osmdkdh@gmail.com (D.-H.K.); gorgeousious@naver.com (S.S.N.); bcchoikr@dsmc.or.kr (B.-C.C.)

**Keywords:** elbow, transolecranon fracture dislocations, olecranon fracture

## Abstract

The purpose of this study is to provide a systematic review of the definition, ideal surgical method, complications, and prognosis of trans-olecranon fracture dislocations. An electronic search was performed in the PubMed, EMBASE, Scopus, and MEDLINE databases. The eligibility criteria included retrospective clinical study and review article in subjects older than 18 years with trans-olecranon fracture dislocations. Trans-olecranon fracture dislocations are defined as fractures in which the stability of the ulnohumeral joint is lost due to the intra-articular fracture of the olecranon without disruption of the proximal radioulnar joint. The seven papers were included that met the eligibility criteria for the quantitative synthesis. Findings indicate that a pre-contoured plate was used in 88.3% of cases (68 of 77 reports), with no reports of complications, suggesting that the pre-contoured 3.5 mm plate is the first choice of treatment. Postoperative mean elbow range of motion for the flexion–extension arc was 121.1° and 146.5° for the pronation-supination arc. Methods for postoperative clinical scores included the Broberg/Morrey rating with a result of excellent or good in 82.9% of cases, the ASES score with a mean of 88.7, and the DASH score with a mean of 11.75. Complications included heterotopic ossification in 21.9% (23/105) of cases, arthrosis in 25.7% (27/105) of cases, nerve damage in 18.1% (19/105) of cases, and osteoarthritis in 14.3% (15/105). With better understanding of the mechanism of injury and proper diagnosis and treatment, findings of the current review suggest a positive outcome. PROSPERO registration No.: CRD42019126568.

## 1. Introduction

The elbow is a complex joint consisting of ulnohumeral, proximal radioulnar, and radiocapitellar articulation. Trans-olecranon fracture dislocations are defined as fractures in which the stability of the ulnohumeral joint is lost due to intra-articular fracture of the olecranon with no disruption of the proximal radioulnar joint [1,2]. In the case of trans-olecranon fracture dislocations, damage is the result of high energy in the mid-range flexion state, causing ulnohumeral joint discontinuity and radiocapitellar dislocation, resulting in radial head anterior displacement relative to the capitellum [3]. Additionally, trans-olecranon fracture dislocations often accompany radial head fractures or coronoid process fractures [2,4,5,6]. Due to challenges with misclassification and the rare occurrence of trans-olecranon fracture dislocations, they were only recently reported by Biga and Thomine when they were distinguished from Monteggia Bado type 1 fractures [7]. Unlike the Monteggia fracture, the trans-olecranon fracture dislocation preserves the proximal radioulnar joint [8,9], and rarely have an accompanying ligament injury, though joint injury is more extensive than in the Monteggia fracture [10].

The goal of surgery in trans-olecranon fracture dislocations is to restore the trochlear notch, whereas, in the case of a Monteggia fracture, emphasis is on the anatomical reduction to align the ulnar diaphyseal fracture [6]. For olecranon fractures, open reduction and internal fixation with plate or tension band wiring have been used; however, discussion of the prognosis is limited and insufficient. Given the relatively rare nature of this fracture, and findings that it is often misdiagnosed, few studies have been conducted. The purpose of the current study is to provide a systematic review of the definition, ideal surgical method, complications, and prognosis of trans-olecranon fracture dislocations.

## 2. Methods

### 2.1. Search Strategy

This study was registered on PROSPERO (CRD42019126568) and followed the Preferred Reporting Items for Systematic Reviews and Meta-analyses (PRISMA) reporting guidelines. Using the PubMed, EMBASE, Scopus, and MEDLINE databases, the search was conducted using the following search terms: “Olecranon fracture dislocations”, “trans-olecranon fracture dislocations”, and “anterior olecranon fracture dislocations”. The referenced papers included in reviewed studies were examined as well. The final search was performed on 10 July 2020.

### 2.2. Eligibility Criteria

Studies that were written in English included the olecranon fracture dislocation, and had the full text available included in the review. Papers that were not written in English did not match the topic, were written exclusively about Monteggia fractures, lacked specifics regarding the population, and were either a case report or review article excluded from the review (Figure 1) [11,12].

### 2.3. Data Extraction and Analysis

All data were extracted in a predetermined format. The first author, title, published journal, year, type of study, design, and level of evidence of each study are summarized in the first data column. The following were extracted for patient demographics: the number of patients, mean age, male:female ratio, dominant limb, and mean follow-up period. Fracture characteristics included injury mechanism, fracture pattern, open fracture, radial head or coronoid process fracture, accompanying injury, and surgical method. Assessment of clinical outcomes included the Broberg/Morrey rating, American Shoulder and Elbow Surgeons (ASES) scores, the Disabilities of the Arm, Shoulder and Hand (DASH) scores, Mayo Elbow Performance Index (MEPI) scores, and the average range of motion of elbow and forearm. Postoperative complications included: heterotopic ossification, arthrosis, nerve injury, osteoarthritis, radioulnar synostosis, persistent pain, loosening, infection, delayed union, nonunion, stiffness, ulnohumeral instability, and malunion. Two reviewers conducted a full-text review blinded to the citation of each paper. The level of evidence was evaluated based on the guidelines of the Oxford Centre for Evidence-Based Medicine.

## 3. Results

Initially, 101 studies were selected according to the selected search terms. Studies were then organized by database and duplicate papers were removed, 39 studies remained and were selected. The abstract for all 39 papers were examined and 19 studies were selected after excluding studies that did not match the topic, did not fit the study type, or were not written in English. Of the 19 that were selected, an additional 12 articles that did not include the population, did not include the treatment options, were not limited to patients with olecranon fracture dislocations, and did not have a full-text available were excluded. The seven remaining papers were selected for systematic review and included in the current study (Figure 1).

All demographic data were collected and averaged. A total of 105 patients were included in the seven papers reviewed [2,5,6,10,13,14,15]. The average age was 42.1 years (range 14–82) and included 69 male and 29 female patients [2,5,6,10,14,15], the dominant limb of the injured one was 41% in three studies [2,5,14], and the mean follow-up period was 68.8 months (Table 1).

Six of the seven studies reported the mechanism of injury as follows (*n* = number of patients): motor vehicle accident (*n* = 26, 53.1%), fall from height (*n* = 9, 18.4%), fall from standing (*n* = 6, 12.2%), roller skating (*n* = 3, 6.1%), assault (*n* = 2, 4.1%), snow mobile accident (*n* = 1, 2%), bicycle accident (*n* = 1, 1.4%), and direct blow (*n* = 1, 2%) [2,5,6,10,13,14]. Five studies reported the fracture patterns as either a simple fracture (*n* = 10, 18.2%) a comminuted fracture (*n* = 45, 81.8%), or an open fracture (*n* = 28, 26.7%) [2,5,6,14,15]. Coronoid process fractures accompanied 53.3% (*n* = 56) of transolecranon fracture dislocations, and radial head fractures accompanied 17.1% (*n* = 18). Seven cases had both a coronoid process fracture and a radial head fracture (*n* = 7, 6.7%) [5,10,15]. Five studies reported that the procedure was performed with a dorsal mid longitudinal approach [2,6,10,14,15]. In the surgical method, open reduction and internal fixation with plate were 77 out of 105 cases, and the following types of plates: dynamic compression plate (*n* = 20, 26%), recon plate (*n* = 13, 16.9%), semi and 1/3 tubular plate (*n* = 9, 11.7%), and an un-specified precontoured olecranon plate (*n* = 35, 45.5 %) were used in 77 cases which used plate fixation. In the case of tension band wiring, there were 16 cases, 15 cases using k-wire, one case using suture, and one case transfixed using k-wire with cast immobilization. Assessments of clinical outcomes used various rating and scoring systems. Six studies used the Broberg/Morrey rating, with 35.7% (*n* = 25) scored as excellent, 47.1% (*n* = 33) as good, 10% (*n* = 7) as fair, and 7.1% (*n* = 5) as poor [2,5,6,13,14,15]. Three studies used the American Shoulder and Elbow Surgeons (ASES) scores, with a mean score of 88.71 [5,6,15], while two studies used the Disabilities of the Arm, Shoulder and Hand (DASH) scores with a mean score of 11.75 [10,13]. One study used the Mayo Elbow Performance Index (MEPI) scores, finding a rating of excellent for 50% (*n* = 5) of cases, good for 30% (*n* = 3), and fair/poor for 10% (*n* = 1) [13]. The postoperative means for elbow range of motion were as follows: flexion was 124°, flexion contracture was 18.7°, extension was −3°, flexion–extension arc was 121.1°, pronation-supination arc was 146.5°, supination was 76.2°, and pronation was 73.4° (Figure 2).

Among complications in all 105 cases, arthrosis was the most common at 25.7% (27/105), followed by heterotopic ossification at 21.9% (23/105) and nerve injury at 18.1% (19/105) (Table 1). The re-operation rate was 29.5% (31/105) and one of the following reasons was carried out: failed tension in six cases, failed 1/3 tubular plate fixation in three cases, elective hardware removal in nine cases, capsular release in three cases, ulnar nerve transposition in one case, and ulnar nonunion in one case (Table 2 and Table 3).

The purpose of the current study was to present a systematic review for diagnosis, treatment, prognosis, and complications of trans-olecranon fracture dislocations. Results indicate that trans-olecranon fracture dislocations have a low frequency of instability and a good prognosis when the trochlear notch is properly restored by rigid plate fixation, contrary to the severity of elbow dislocation and peripheral fractures due to high energy trauma.

In the reviewed cases, the proportion of comminuted fractures was 81.8% (45/55) and the open fracture rate was 26.7% (28/105), supporting high energy trauma as a source of trans-olecranon fractures. Coronoid process fractures accounted for 53.3% (56/105), radial head fractures for 17.1% (18/105), and 6.7% (7/105) of fractures were both types. The reason the frequency of the coronoid process fracture is relatively high is thought to reflect the characteristics of fractures where disruption of the ulnohumeral joint and radiocapitellar dislocation both occur. The most frequently reported complications were arthrosis and heterotopic ossification, which likely reflect the characteristics of complex fracture dislocation. Conversely, the rate of limitation of motion was 1.9% (2/105), allowing for joint exercise soon after surgery.

Trans-olecranon fracture dislocations are defined as the anterior dislocation of the elbow associated with an olecranon fracture. The injury mechanism is often high energy trauma, and the collateral ligament is frequently spared (3.8% of 105 cases). According to Haller et al., trans-olecranon fractures are rare enough to account for only 6% of proximal ulnar fractures, which is likely the reason for a lack of publications aside from one by Ring published in 1998 [2,10]. Trans-olecranon fracture dislocations are often misdiagnosed as anterior Monteggia fractures; however, a key difference is that the radioulnar association is maintained in trans-olecranon fracture dislocations [2]. Furthermore, trans-olecranon fracture dislocations have been confused with Monteggia fracture Bado type I which refers to the fracture and angular deformity in the ulnar diaphysis, and anterior dislocation of the radial head due to radioulnar joint dissociation.

In the case of comminuted fractures where anatomical reduction is difficult to complete, it is important to restore the trochlea groove by restoration of the trochlear notch [6]. In Monteggia fracture Bado type I, the anatomical reduction of the proximal ulna is the focus, while, in other complex elbow fracture dislocations, it is important to correct the instability due to damage of the annular or collateral ligaments. Conversely, in trans-olecranon fracture dislocations, the stable restoration of the greater sigmoid notch is an issue [1,3,6]. Scolaro et al. suggested that, in addition to attention of the trochlear notch, anatomical reduction of ulnar is equally important [1].

In the current review, trans-olecranon fracture dislocations were often accompanied by coronoid process fractures and radial head fractures. For stable fixation of coronoid process fractures, Ring et al. suggested that interfragmentary screw fixation was necessary, while other papers did not provide further explanations of the treatment [2]. In the case of radial head fracture, excision was performed when the fragment was marginal, and radial head replacement was performed when comminution was severe [2,6,10].

In the study by Haller et al., four cases out of 35 cases of lateral ulnar collateral ligament (LUCL) injury were reported, all of which were accompanied by both coronoid process and radial head fracture. Considering that there was no LUCL injury in previous research, it is likely that there was only one case of both coronoid process and radial head fracture in other studies as well [5,10]. Öztürkmen et al. reported two cases of repair due to damage to the lateral collateral ligament, though the direction of fracture dislocation was not indicated, excluding the cases from analysis [15]. In two cases, ulnohumeral instability was reported, but this was due to failed anatomical reduction as both cases were treated conservatively using a long arm cast rather than surgical intervention [6].

Ring et al. recommended a limited contact-dynamic compression plate (LC-DCP) as the tubular plate is less rigid with a lower possibility of cyclic fatigue, suggesting that tension band wiring could be used in simple fractures [2]. However, Mortazavi et al. reported that tension band wiring is not recommended due to a lack of stability [5]. In the current review, revision surgery due to loosening or fixation failure was reported in six cases in open reduction and internal fixation with k-wire and in three cases using a 1/3 tubular plate [2,5,6,13,14]. The pre-contoured plate was used in 68 cases (68/77, 88.3%), making it the most frequently used, and revision surgery was not needed, suggesting that the pre-contoured 3.5 mm plate is the favored choice for treatment. In most cases of trans-olecranon fracture dislocations, the probability of elbow instability is very low and early active range of motion is required for a good postoperative prognosis [14]. Morrey et al. found that coronoid process fractures of more than 50% are the cause of traumatic arthrosis, though Ring et al. said that such cases do not always lead to poor results [2,16]. The prevalence of heterotopic ossification was 21.9% (23/105) overall in this systematic review but was reported as 42.9% (15/35) in a previous paper by Haller et al. Haller reported that the high prevalence is due to the lack of nonsteroidal anti-inflammatory medications and the absence of radiation prophylaxis to reduce the possibility of postoperative nonunion [10]. Haller et al. also reported that the incidence of post-traumatic osteoarthritis was statistically significant in cases with concomitant injury (radial head fracture, coronoid process fracture, capitellum fracture, and/or ligament injury) but not in olecranon fracture only [10]. Postoperative clinical scores included the Broberg/Morrey rating, which had a mean rating of excellent or good at 82.9%, while the mean ASES score was 88.7, and the mean DASH score was 11.75, indicating a good prognosis when the mechanism of fracture dislocations was high energy trauma. Postoperative mean elbow ROM obtained good results with rapid rehabilitation after the operation in most of the seven papers.

There are some limitations to this study that should be considered. First, most of the studies that extracted the data had a low level of evidence. Additionally, there are limitations due to the quality of exclusion criteria, reported preoperative factors, limited long-term follow-up, and a lack of randomization or blinding to avoid bias. In the current systematic review, there were no randomized clinical trials or case–control studies, and only case series or cohort studies were included. Therefore, integrated analysis was not possible, and strong conclusions could not be drawn. Though meta-analysis was attempted when collecting data on posterior olecranon fracture dislocations, it was not possible as cases of Monteggia type IIb fracture were included, resulting in a high possibility that bias may be accompanied in determining the range.

## 4. Conclusions

The trans-olecranon fracture dislocations have a low frequency of instability and a good prognosis when the trochlear notch is properly restored by rigid plate fixation, contrary to the severity of elbow dislocation and peripheral fractures due to high energy trauma. 

## Figures and Tables

**Figure 1 diagnostics-10-01058-f001:**
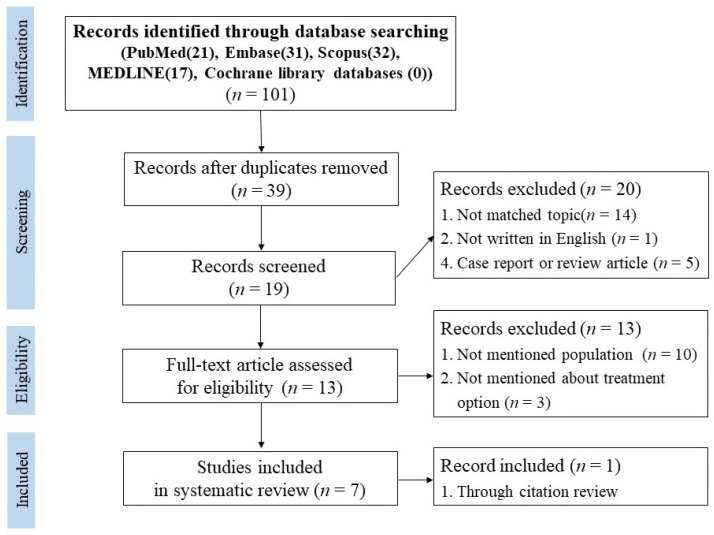
Preferred reporting items for systematic reviews.

**Figure 2 diagnostics-10-01058-f002:**
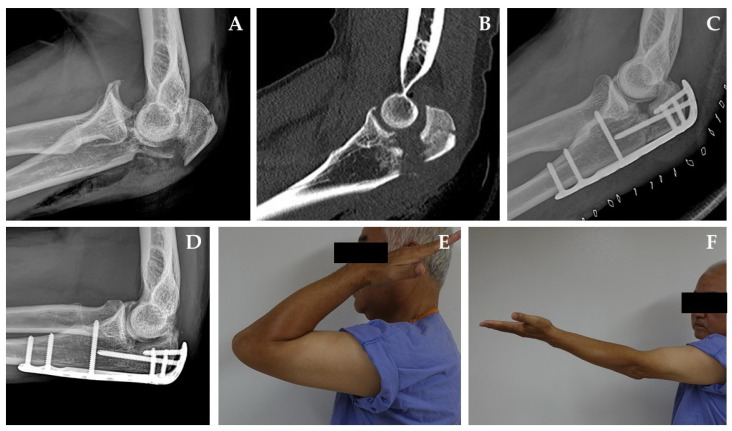
A sixty-year-old man sustained a trans-olecranon fracture-dislocation of the left elbow due to a two-meter fall. (**A**) The initial lateral radiograph revealed a complex fracture pattern of olecranon and the forearm was dislocated anteriorly. (**B**) There were impacted articular fragments at the center of olecranon fossa. (**C**) Lateral radiograph taken two days after the operation demonstrates restoration of the normal ulnotrochlear relationship with a minimal bone defect of the bare area of olecranon. (**D**) A lateral radiograph taken ten months after the operation shows the union of fracture sites without any arthritic change. (**E**,**F**) A good functional result was achieved.

**Table 1 diagnostics-10-01058-t001:** Patient demographics.

Authors	Number of Patients	Mean Age	Male	Female	Dominant Limb Ratio (%)	Mean Follow Up Period (month)
Ring et al. [2]	17	38	14	3	41.2	25
Doornberg et al. [6]	10	49	7	3	N/R	79.2
Moushine et al. [14]	14	54	6	8	50	42
Mortazavi et al. [5]	8	35	7	1	25	37.4
Haller et al. [10]	35	45	25	10	N/R	28
Lindenhovius et al. [13]	10	30	N/R	N/R	N/R	235.2
Öztürkmen et al. [15]	11	44	7	4	N/R	34.9
	105	42.1	66	29	41	68.8

**Table 2 diagnostics-10-01058-t002:** Characteristic of fracture mechanism, treatment, and clinical score outcomes.

Authors	Ring et al. [2]	Doornberg et al. [6]	Moushine et al. [14]	Mortazavi et al. [5]	Haller et al. [10]	Lindenhovius et al. [13]	Öztürkmen et al. [15]
Number	17	10	14	8	35	10	11
Mechanism of injury							
MVA *	11	4	6	5	12	N/R	2
Fall from height	3	2	3	1	0	N/R	4
Fall from standing height	1	2	2	1	8	N/R	3
Assault	2	0	0	0	N/R	N/R	0
Bicycle accident	0	1	0	0	N/R	N/R	0
Roller skating accident	0	0	3	0	N/R	N/R	0
Direct blow	0	0	0	1	N/R	N/R	1
Sports injury	0	0	0	0	N/R	N/R	1
Snow mobile accident	0	1	0	0	N/R	N/R	0
Fracture patterns							
Simple, oblique	3	1	1	1	N/R	N/R	1
Simple, transverse	0	0	3	0	N/R	N/R	0
Complex, comminuted	14	9	10	7	N/R	N/R	5
Open fractures							
Total	5	3	6	1	12	1	1
GA I	0	0	0	1	3	1	1
GA II	2	2	4	0	7	0	0
GA IIIA	2	1	0	0	2	0	0
GA IIIB	1	0	2	0	0	0	0
Coronoid process fractures	8	5	5	4	23	5	6
Radial head fractures	2	1	1	2	9	0	3
Accompanied injury	7: segmental ulnar fx. *	0	N/R	1: humerus neck fx. 1: calcaneus fx.	2: capitellum fx. 1: lateral condylar fx. 1: compartment syndrome	N/R	N/R
Approach	Dorsal mid longitudinal	Dorsal mid longitudinal	Dorsal mid longitudinal	N/R	Dorsal mid longitudinal	N/R	Dorsal mid longitudinal
Surgical method							
ORIF * c plate							
3.5 mm Limited contact DCP *	9	5	0	0	0	0	N/R
3.5 mm DCP	0	1	1	0	0	4	N/R
3.5 mm Recon plate	2	0	4	7	0	0	N/R
Semitubular plate	2	0	0	0	0	0	N/R
1/3 tubular plate	2	1	2	0	0	2	N/R
3.5 precontoured plate	0	0	0	0	35	0	N/R
ORIF c tension band	2	2	7	1	0	3	N/R
ORIF c tension suture	0	1	0	0	0	0	N/R
RC * transfixation c wire	0	0	0	0	0	1	N/R
Clinical Scores							
Broberg/Morrey rating	Excellent: 7 Good: 8 Fair: 2 Poor: 0	Excellent: 4 Good: 5 Fair: 0 Poor: 1	Excellent: 4 Good: 6 Fair: 2 Poor: 2	Excellent: 2 Good: 5 Fair: 1 Poor: 0	N/R	Excellent: 5 Good: 3 Fair: 0 Poor: 2	Excellent: 3 Good: 6 Fair: 2 Poor: 0
ASES *	N/R	89.2	N/R	89	N/R	N/R	87.8
DASH *	N/R	N/R	N/R	N/R	9 (28 patients)	14.5	N/R
MEPI *	N/R	N/R	N/R	N/R	N/R	Excellent: 5 Good: 3 Fair: 1 Poor: 1	N/R
Average ROM *							
Flexion	127°	130°	125° (110–140°)	115° (85–140°)	123° (45–145°)	N/R	N/R
Extension	N/R	N/R	−22° (−40–0°)	N/R	16° (0–80°)	N/R	N/R
Flexion–extension arc	N/R	110°	N/R	N/R	107° (10– 130°)	Early F/U *: 117° (70–140°) Long term F/U: 124° (50–145°)	107.3°
Flexion contracture	14°	20°	N/R	22° (0–45°)	N/R	N/R	N/R
Supination	Normal	75°	76° (60–90°)	83° (80–85°)	77° (20–85°)	N/R	69.1°
Pronation	Normal	80°	68° (55–85°)	75° (40–90°)	65° (30–70°)	N/R	79.1°
Supination-pronation arc	N/R	155°	N/R	157.5° (120–173°)	137° (50–155°, 29 pt.)	Early F/U: 123° (0–180°) Long term F/U: 133° (0–170°)	150°

*: fx.: fracture; RC: Radiocapitellar; MVA: motor vehicle accidents; ROM: range of motion; ASES: American shoulder and elbow surgeons; DASH: Disabilities of the Arm Shoulder and Hand; MEPI: Mayo Elbow Performance Index; GA: Gustilo Anderson; ORIF: open reduction and internal fixation; DCP: dynamic compression plate; NR: not recorded.

**Table 3 diagnostics-10-01058-t003:** Complications about trans-olecranon fracture dislocations.

Authors	No *	HO *	Arthrosis	Nerve Injury	Osteoarthritis	Radioulnar Synostosis	Persistent Pain	Loosening	Infection	Delayed Union	Nonunion	Limitation of Motion	UH * Instability	Malunion	Re-Operation
Ring D et al. [2]	17	1	2	0	N/R *	N/R	3	2: 1/3 tubular	N/R	N/R	0	1	0	0	2: 1/3 tubular plate loosening
Doornberg et al. [6]	10	1	2	1 BPI *, 1 ulnar nerve	N/R	4		1: tension band wire	N/R	N/R	N/R	1 (MUA *)	2	2	5: 1 failed tension band3: capsular release1: ulnar nonunion
Moushine et al. [14]	14	4	N/R	0	4	N/R	N/R	N/R	0	3	2	N/R	0	N/R	4: tension band wiring failed
Mortazavi et al. [5]	8	1	N/R	N/R	0	N/R	1	1: tension band	N/R	N/R	1	1	N/R	N/R	4: 1 failed tension band, 3: elective hardware removal
Haller et al. [10]	35	15	13	6: ulnar nerve	11	2	2	N/R	4 (I & D with IV * Anti-biotics)	N/R	N/R	N/R	N/R	N/R	14: 4 devices remove
Lindenhovius et al. [13]	10	N/R	5	7: ulnar nerve	N/R	N/R	N/R	N/R	N/R	N/R	N/R	N/R	N/R	N/R	2: 1/3 tubular plate loosening
Öztürkmen et al. [15]	11	1	5	1: ulnar nerve	N/R	N/R	N/R	N/R	N/R	N/R	N/R	N/R	N/R	N/R	N/R

*: No: number of patients; HO: Heterotopic ossification; pt.: patients; UH: ulnohumeral; IV: intravenous; NR: not recorded; POD: postoperative day; DCP: dynamic compression plate; AIBG: autogenous iliac bone graft; BPI: brachial plexus injury; MUA: manipulation under anesthesia4. Discussion.

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
