# Peer review of "Trans-Olecranon Fracture-Dislocations of the Elbow: A Systematic Review"

_diagnostics, 2020, doi:10.3390/diagnostics10121058_

Round 1
Reviewer 1 Report
This is a systematic review of the treatment of transarticular olecranon fractures. It comprises of in the end seven articles, with make use in majority of the 3.5 pre contoured specialty plates. There was a high proportion of comminuted and open fractures, indicating the high energy type of fractures.
The guidelines to perform a systematic review are followed. A myriad of techniques were used, not easily comparable or to be combined in a basic comparable dataset. Moreover the number of patients in the ultimately evaluated studies were low, between 17 and 35.
Major complications were arthrosis and heterotopic ossification both in a quarter of the patients. The reported re-operation was needed in almost 1/3 of the patients.
This is a straight forward systematic review, timely, adequately performed and with only limited amount of patients in the various studies. The discussion is adequate, to the point and with all relevant aspects.
Author Response
Point 1: This is a systematic review of the treatment of transarticular olecranon fractures. It comprises of in the end seven articles, with make use in majority of the 3.5 pre contoured specialty plates. There was a high proportion of comminuted and open fractures, indicating the high energy type of fractures.
The guidelines to perform a systematic review are followed. A myriad of techniques were used, not easily comparable or to be combined in a basic comparable dataset. Moreover the number of patients in the ultimately evaluated studies were low, between 17 and 35.
Major complications were arthrosis and heterotopic ossification both in a quarter of the patients. The reported re-operation was needed in almost 1/3 of the patients.
This is a straightforward systematic review, timely, adequately performed and with only limited amount of patients in the various studies. The discussion is adequate, to the point and with all relevant aspects.
Response 1: Thank you for your comment. Although the authors wrote a review article with few papers, I think that it can gain general knowledge about this injury and improve the quality of the patient's treatment.
Reviewer 2 Report
Dear Authors,
Thank you for the opportunity to review your work.
I think that it is a well conducted and well written systematic review on a rare type of fractures and limited reported series.
Just few suggestions:
Maybe it could be useful, in order to clarify the definition of transolecranon fracture dislocations, to move pag. 1 lines 37-41 at ilne 31 just after ref [1-2].
Pag. 8 line 171 "considering that no LUCL was reported": have you missed something? please clarify.
Pag. 8 line 179: please change "Mortarazi" in Mortazavi
Pag. 9 line 198: please clarify "directed"
I think that it could be useful to add a picture/s in order to show the fracture that you have written about.
Author Response
Point 1: Maybe it could be useful, in order to clarify the definition of transolecranon fracture dislocations, to move pag. 1 lines 37-41 at ilne 31 just after ref [1-2].
Response 1: Thank you. I changed that according your comment.
Point 2: Pag. 8 line 171 "considering that no LUCL was reported": have you missed something? please clarify.
Response 2: Thank you. I revised that.
Point 3: Pag. 8 line 179: please change "Mortarazi" in Mortazavi
Response 1: Thank you. I changed that.
Point 4: Pag. 9 line 198: please clarify "directed"
Response 4: Thank you. I revised that.
Point 5: I think that it could be useful to add a picture/s in order to show the fracture that you have written about.
Response 5: Thank you for your comment. I added it to Figure 2 using the case of our institution related to the above disease.